# Unusual bromine enrichment in the gastric mill and setae of the hadal amphipod *Hirondellea gigas*

**Satoshi Okada**[1]*, **Chong Chen**[1], **Hiromi Kayama Watanabe**[1], **Noriyuki Isobe**[2], **Ken Takai**[1]

**1** Institute for Extra-cutting-edge Science and Technology Avant-garde Research (X-STAR), Japan Agency for Marine-Earth Science and Technology (JAMSTEC), Yokosuka, Kanagawa, Japan, **2** Biogeochemistry Research Center, Research Institute for Marine Resources Utilization (MRU), Japan Agency for Marine-Earth Science and Technology (JAMSTEC), Yokosuka, Kanagawa, Japan

* okadasa@jamstec.go.jp

## Abstract

The hadal amphipod *Hirondellea gigas* is an emblematic animal of the Pacific trenches, and has a number of special adaptations to thrive in this 'extreme' environment, which includes the deepest part of the Earth's ocean. One such adaptation that has been suggested is the presence of an 'aluminum gel shield' on the surface of its body in order to prevent the dissolution of calcitic exoskeleton below the carbonate compensation depth. However, this has not been investigated under experimental conditions that sufficiently prevent aluminum artefacts, and the possibility of other elements with similar characteristic X-ray energy as aluminum (such as bromine) has not been considered. Here, we show with new electron microscopy data gathered under optimized conditions to minimize aluminum artefacts that *H. gigas* actually does not have an aluminum shield–instead many parts of its body are enriched in bromine, particularly gastric ossicles and setae. Results from elemental analyses pointed to the use of calcite partially substituted with magnesium by *H. gigas* in its exoskeleton, in order to suppress dissolution. Our results exemplify the necessity of careful sample preparation and analysis of the signals in energy-dispersive X-ray spectroscopic analysis, and the importance of analyses at different electron energies.

## Introduction

Many animal groups have evolved hard structures for skeletal support (e.g. vertebrate bones), protection (e.g. molluscan shells), and feeding (e.g. mouth parts and teeth), among other functions–and often for multiple purposes. These structures are composed of either inorganic minerals, such as calcite, aragonite, and hydroxyapatite, or organic polymers, such as chitin, or a mixture of both. Inorganic biomineralized parts exposed to the external environment are often coated with organic matrices, such as molluscan shells covered by a layer of periostracum made of a combination of chitin and proteins [1]. In some cases, such as the marine polychaete worm *Nereis virens*, heavy transition metals and halogens are utilized to coat the mineralized

**Data Availability Statement:** All relevant data are within the paper and its Supporting Information files.

**Funding:** S.O., [Grant-in-Aid for Young Scientists Grant Number JP19K15379], Japan Society for the

Promotion of Science (JSPS), https://www.jsps.go.jp/; C.C. and H.K.W., [Grant-in-Aid for Scientific Research Grant Number 18K06401], JSPS. The funders had no role in study design, data collection and analysis, decision to publish, or preparation of the manuscript.

**Competing interests:** The authors have declared that no competing interests exist.

tissues [2,3]. These external layers often play roles in the secretion of the biominerals, and they can also reinforce the structure and prevent the deterioration of the underlying mineral layer. For example, bivalves living deeper than the carbonate compensation depth (CCD; typically ~4500 m), where calcium carbonate dissolves quickly, use their periostracum to buffer against natural shell dissolution [4].

Crustaceans are among the most successful invertebrates in marine environments and use hard structures for both protection in the form of calcified exoskeletons and feeding in the form of gastric mills. Their exoskeletons are mainly composed of chitinous biopolymers hardened through sclerotization, the cross-linking of protein components [5,6]. To achieve further mechanical strength and/or adaptive functionality, the exoskeletons are often additionally impregnated with other materials. The most widespread example is amorphous calcium carbonate and calcium phosphate [5–7]. Although most crustaceans with calcified exoskeleton predominantly use calcium carbonate, some such as barnacles in the genus *Ibla* chiefly utilize calcium phosphate [8,9]. Phosphate is more common in crustacean mandibles [10]. Calcium phosphate is often used to strengthen the mandible, such as in the crayfish which forms fluoroapatite as a crown on its mandible [11,12]. Some crustaceans, such as woodlice, use crystalline magnesium calcite where a part of calcium is substituted with magnesium [13]. Furthermore, a number of crustaceans exhibit further coatings outside the exoskeleton. For examples barnacles overlay the hardened plates with cuticular and epicuticular layers [8], deep-sea ostracods cover the mineralized layer with epidermis to prevent mineral dissolution even below the CCD [14,15], and crabs such as *Pachygrapsus crassipes* possess bromine-enriched coating on their claw tips that increases the fracture-resistance [16,17].

The hadal amphipod *Hirondellea gigas* is one of the most numerically dominant crustacean macrofauna in the deepest areas of the ocean, including the Challenger Deep in the Mariana Trench (10,994 m) [18]. As special adaptations to this extreme hadal environment, this species possesses unique cellulase and is alleged to digest polysaccharides from sunken plant matter using a unique cellulase[18,19]. Perhaps most strikingly, it is said to secrete aluminum to the outside of its exoskeleton to prevent the dissolution of calcium-based minerals, dubbed the 'aluminum shield' [20]. Based on the combination of energy-dispersive X-ray analysis (EDS) under scanning electron microscopy (SEM), and aluminum extraction experiments from the sediment of Challenger Deep, Kobayashi et al. [20] expected that the 'aluminum shield' of *H. gigas* would be in gel form. However, their results were inconclusive and ambiguous due to the localization pattern of aluminum, where the signal intensity of aluminum was especially high only at the edge of the samples close to the aluminum stub most prone to artefacts. Furthermore, samples were not covered with conductive coatings for imaging and EDS analyses to prevent the sample from charging-up, despite observations being undertaken in the aluminum-rich environment that is the SEM chamber. In addition, EDS spectra under a scanning transmission electron microscope (STEM) of the exoskeleton did not provide enough information to support the existence of aluminum and its detailed nanostructure [20].

There are two possibilities where aluminum artefacts may be observed: 1) backscattered electrons (BSE) from the observed specimen hitting the SEM chamber made of aluminum alloy to generate secondary electrons (called SE3) as well as characteristic X-ray of aluminum [21], 2) overlapping of other elements that have similar characteristic X-ray energy signatures to that of aluminum (Al K-line, 1.486 keV) and which are inseparable using conventional EDS whose energy resolution is ~130 eV. These elements include bromine (Br L-line, 1.480 keV), thulium (Tm M-line, 1.462 keV), and ytterbium (Yb M-line, 1.521 keV) [22], the latter two are unused in animals [23,24]. In order to confirm the presence of the 'aluminum shield' of *H. gigas* and to further substantiate its distribution and potential function, we designed a more careful experiments to exclude aluminum artefacts. Here, we report our findings from detailed

electron microscopy observations of *H. gigas*, revealing the absence of aluminum on the exoskeleton and instead the accumulation of bromine on hard tissues as the sclerotized spines of its gastric mill components, a rare example of preferentially secreted bromine in the hard structures of animals.

## Materials & methods

### Materials

All reagents were purchased either from Nacalai tesque Inc., Fuji Film Wako Pure Chemical Co., or Kanto Chemical Co., Inc. and used as received. Deionized water was supplied from Merck Milli-Q Integral 5.

### Amphipod samples

*Hirondellea gigas* individuals were collected using a bait trap attached to a Free Fall Camera System at Izu-Ogasawara Trench near the Boso Triple Junction (34˚20.4296'N 142˚0.6006'E, Fig 1A), depth 9218 m on September 4, 2019, on R/V Yokosuka cruise YK19-11. Over 200 individuals (Fig 1B) were collected, of which three individuals were used for this study. The amphipods were immediately immersed in slush nitrogen (ca. –210˚C) for more than a minute to rapidly freeze the exoskeleton and stored at –80˚C during the cruise, then transferred to liquid nitrogen storage (actual storage temperature <–160˚C) after cruise.

### Electron microscope instruments

SEM and STEM imaging were performed on a Helios G4 UX (Thermo Fisher Scientific), equipped with an Octane Elite Super energy dispersive X-ray analysis (EDS) detector (Octane

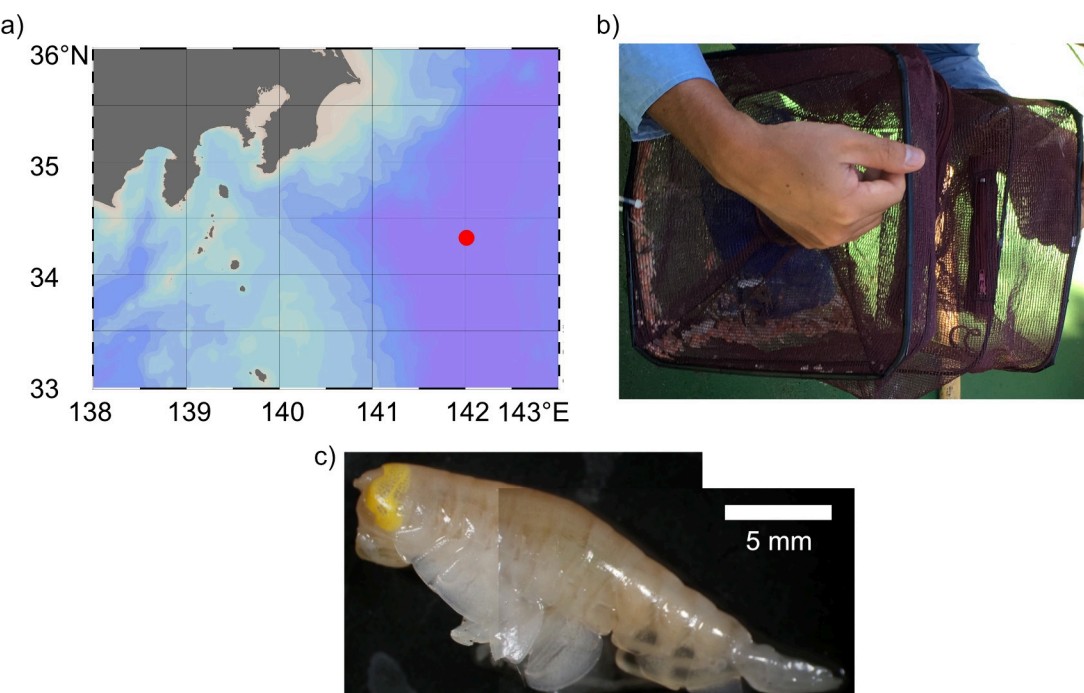

**Fig 1. Sampling of *Hirondellea gigas*.** a) Sampling location represented by the red dot. b) *H. gigas* collected in the bait trap. More than 200 individuals were collected. c) A micrographic montage of *H. gigas* under stereomicroscopy.

Elite Super, AMETEK) and a cryogenic stage with preparation chamber (PP3010T, Quorum). Transmission electron microscopy (TEM) was performed on a Tecnai G2 20 (Thermo Fisher Scientific, operated at 200 kV) equipped with a bottom-mounted 2k × 2k Eagle Charge-Coupled Device camera (Eagle, Thermo Fisher Scientific) and RTEM-S 61700NE an EDS detector (RTEM-S 61700NE, AMETEK). Electron probe microanalysis (EPMA) was performed on a JXA-8500 (JEOL Ltd.) equipped with five wavelength dispersive X-ray detectors (WDS, XM-86030 and XM-86010, JEOL Ltd.).

## Sample preparation and semi-thin section imaging

Frozen amphipods were dehydrated in acetone at –80˚C for three days and gradually warmed to room temperature. Dehydrated amphipods were immersed into 1-*n*-butoxy-2,3-epoxypropane (QY-1, Ohkenshoji Inc.), QY-1/epoxy mixture, then placed in an epoxy resin (TAAB), trimmed into pieces using rasor blades, and embedded in the epoxy resin following the Luft's method [25]. Transverse sections of the head part were taken from anterior to posterior into 1 μm-thick semi-thin sections using an ultramicrotome (Ultracut S or EM UC7, Leica) with a diamond knife (45˚, Diatome). Sections were collected using a nickel loop and placed on a glass slide covered with Kapton tape (Nitto Denko Co., P-224, 64 μm thickness). Slides were then coated with carbon for conductivity using CADE (Meiwaforsis Inc.). Semi-thin sections of pereonite were prepared in the same manner.

Semi-thin sections were subjected to SEM/EDS analysis operated at 1 kV for morphological imaging, and at 10 kV for elemental analysis and mapping. Chlorine was observed uniformly in the epoxy resin as its contaminant, and carbon coating was performed for conductivity on the polyimide-based substrate. However, the localization of these two elements is not discussed in detail.

Elemental analysis at high energy resolution was performed on an EPMA operated at 20 kV and beam current of 12 nA. The elemental colocalization in the EPMA map was quantified using cross-correlation image analysis [26]. The cross-correlation factor g between two elemental maps *a* and *b* was calculated as

$$\gamma(a, b) = \frac{\sum_{ij}\{[I_a(r_{ij}) - \bar{I}_a] \cdot [I_b(r_{ij}) - \bar{I}_b]\}}{\sqrt{\sum_{ij}[I_a(r_{ij}) - \bar{I}_a]^2} \cdot \sqrt{\sum_{ij}[I_b(r_{ij}) - \bar{I}_b]^2}} \tag{1}$$

where $I_x(r_{ij})$ denotes the X-ray intensity of map x at the pixel position (i, j), and $\bar{I}_x$ is the averaged intensity of map x over the image. The cross-correlation calculation and EPMA elemental mapping were implemented in Python 3.9.1.

## Microscopic observation and image analysis of a dissected amphipod

Another frozen individual of *H. gigas* (total length 26.0 mm, telson length 5.3 mm) was defrosted and dissected for three-dimensional structural investigation under a stereomicroscope (SZX7, Olympus) and photos were taken using an Olympus E-M5 Mark III DSLR camera (Fig 1C). The amphipod was dissected into different parts, with observations focusing on: mandible, maxilla, maxilliped, antennae, gastric mill (pterocardiac ossicles in the cardiac stomach), pereonites, pleopod, uropod, and coxal plates. Dissected samples were placed within Eppendorf tubes (0.2 mL), dehydrated by an increasing series of ethanol (50, 60, 70, 70, 70, 80, 90, 99, 100, 100, 100%, with more than 10 minutes at each step), then substituted with *tert*-butanol, frozen at –20˚C, and dried under vacuum ($<1 \times 10^{-1}$ Pa) to obtain dried specimens, which were then mounted on a Kapton tape on a glass slide using double-sided carbon

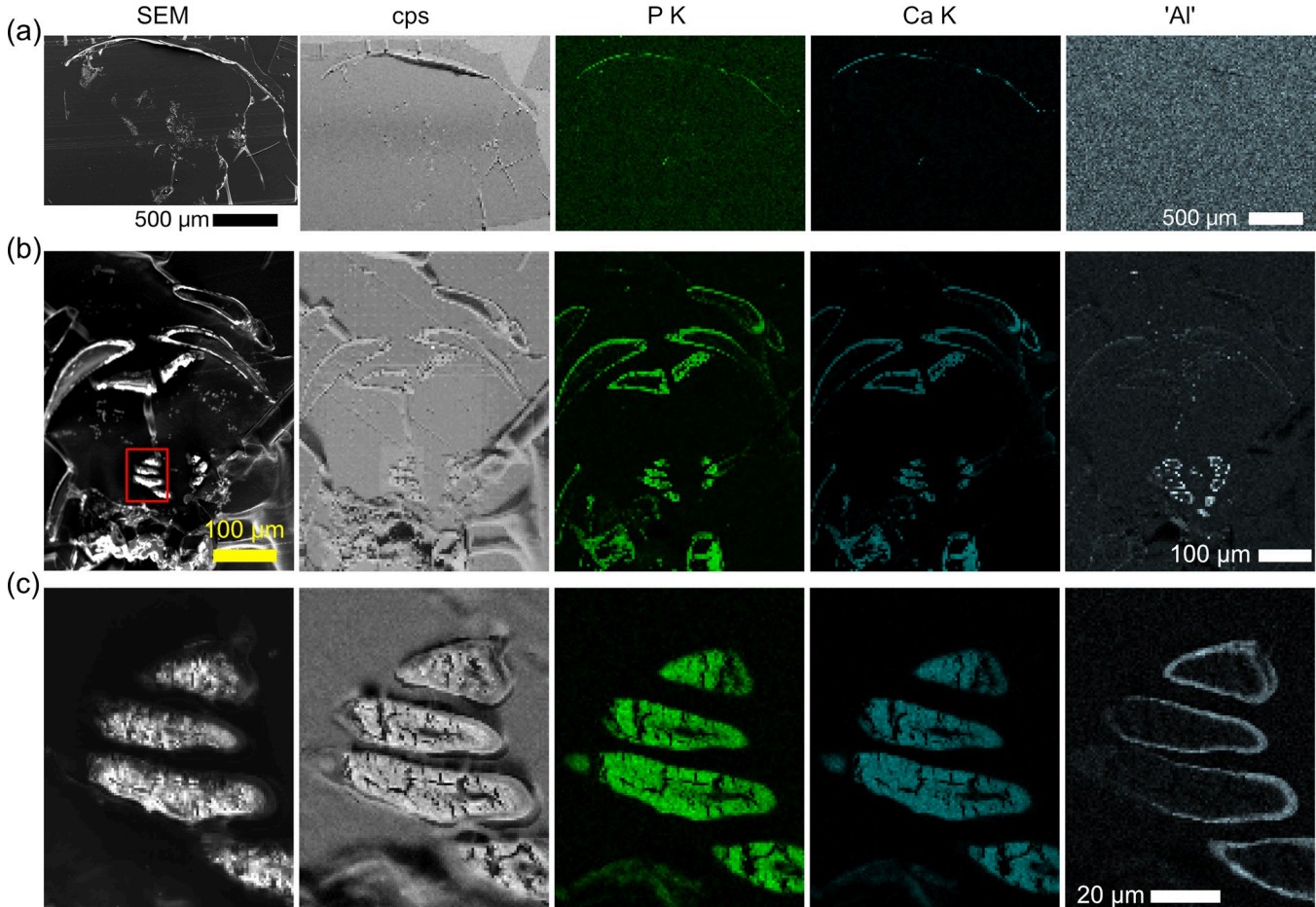

**Fig 2. EDS maps of semi-thin sections of *H. gigas*.** a) Transverse section through a subsample of pereonite. b) Transverse section of the cephalic region showing the gastric mill. c) Magnified images of the gastric mill, the red rectangle area in (b). Images from the left are SEM images at (a,b) 1 kV and (c) 10 kV, total X-ray counts (grayscale), phosphorous (green), calcium (blue), and aluminum-like signals around 1.48 keV (silver). EDS spectra were acquired at 10 kV, thus the K-line of bromine was not observed.

adhesive tape. The mounted samples were coated with carbon for conductivity, and analyzed in SEM/EDS operated at 1 kV for morphological imaging and 10 or 20 kV for EDS analyses.

## X-ray measurement

Dried pereonites were cut into flat pieces and mounted on an X-ray diffractometer (X'Pert PRO, PANalytical) with a Cu radiation source (K$\alpha$ = 1.5418 Å) at 45 kV and 40 mA. The diffractogram from 10° to 60° was acquired at 0.0167° steps at the scan speed of 0.07°/s. The 16 spectra were integrated, and the mineral assignment was performed on the software (High-Score 3.0.5, PANalytical).

## Results

### Elemental distribution and composition of *H. gigas*

To observe the potential aluminum layer in the exoskeleton of *Hirondellea gigas*, the elemental composition was investigated by SEM/EDS of thin sections to avoid artificial aluminum signals caused by the substrate and non-flat surfaces (Figs 2 and S1). EDS maps of the pereonite and

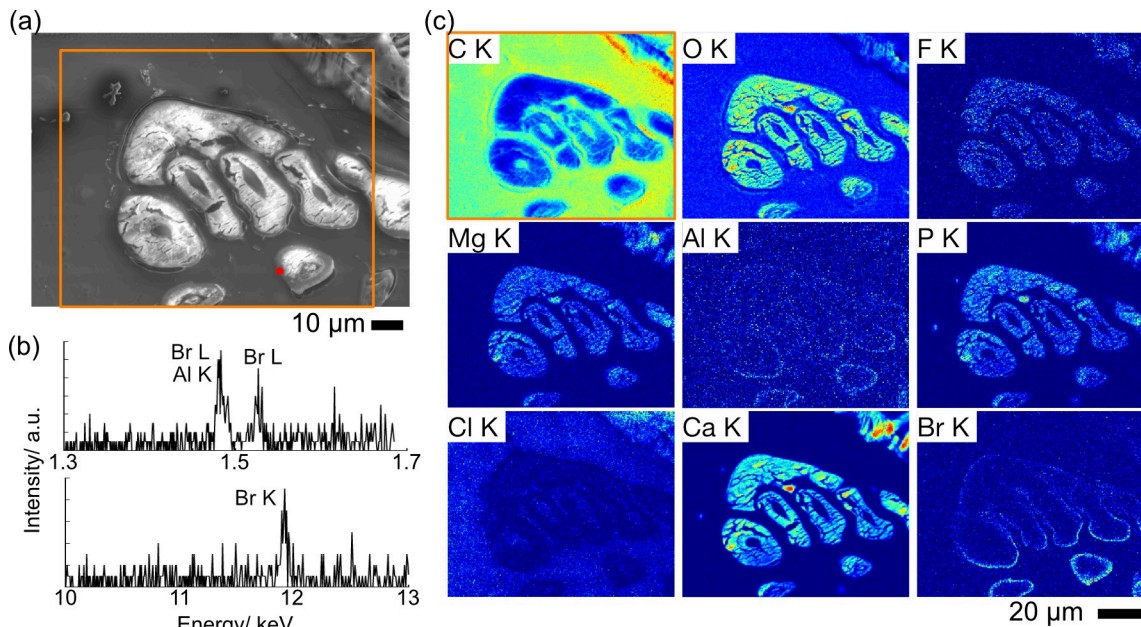

**Fig 3. X-ray spectra and elemental maps of a semi-thin section of the gastric mill in *Hirondellea gigas*.** a) SEM image of the mapped area. b) Point analysis at the red point in (a). c) WDS elemental maps of the orange rectangle in (a). Note that Al K contrast is significantly weaker than that of Br K.

telson showed no evidence of aluminum, while strong signals of carbon, oxygen, phosphorous, and calcium as well as a weak signal of sulfur were observed in the area where bright contrast was observed in the SEM images (Figs 2A and S1A). Other elements, such as sodium, aluminum, and silicon were within the background levels of the continuous X-ray (bremsstrahlung), while chlorine was from the epoxy resin. The EDS maps showed uniform existence of these elements, indicating the compositional homogeneity of the exoskeleton. Soft parts within the exoskeleton were composed of organic materials, and the signal of chlorine corresponded to the epoxy resin. The intensities of the aluminum-like signal (as aluminum overlaps with that of bromine, thulium, and ytterbium) was similar to that of the background signals, indicating that this signal was below the detection limit of the imaging condition used here.

To confirm that the mineral components of the pereonites of *H. gigas* are the same as previously observed (i.e. mainly calcitic) [20], XRD measurements were performed using the dissected dried pereonite. Nine of eleven peaks were assigned to that of calcite (S2 Fig). The residual peaks at 19.7˚ and 26.7˚ were explained by α-chitin, which is ubiquitous in crustacean exoskeletons [27,28]. This result combined with EDS analysis confirmed that the exoskeleton of *H. gigas* is mainly made of calcite and organic matrices, including phosphorous.

The elemental composition of the semi-thin sections through the gastric mill was different from that of pereonite (Figs 2B, 2C, S1B and S1C). The tissues surrounding the gastric mill showed similar elemental content to that of the pereonite. On the other hand, a strikingly different composition was found on the gastric mill itself (Fig 2C) [29] which appeared to form a bilayered structure with an inner layer composed of carbon, nitrogen, oxygen, sodium, magnesium, and phosphorous, while the outer layer was composed of carbon, nitrogen, oxygen, sodium, calcium, and an aluminum-like signal at 1.48 keV. Magnified EDS maps showed the thickness of the outer layer to be ~1 μm.

EPMA analysis was also performed using similar sections of *H. gigas* (Fig 3A) to correctly assign the X-ray signals at 1.48 keV. Qualitative WDS spectrum of the outer layer of the gastric

(a)

(b)

(c)

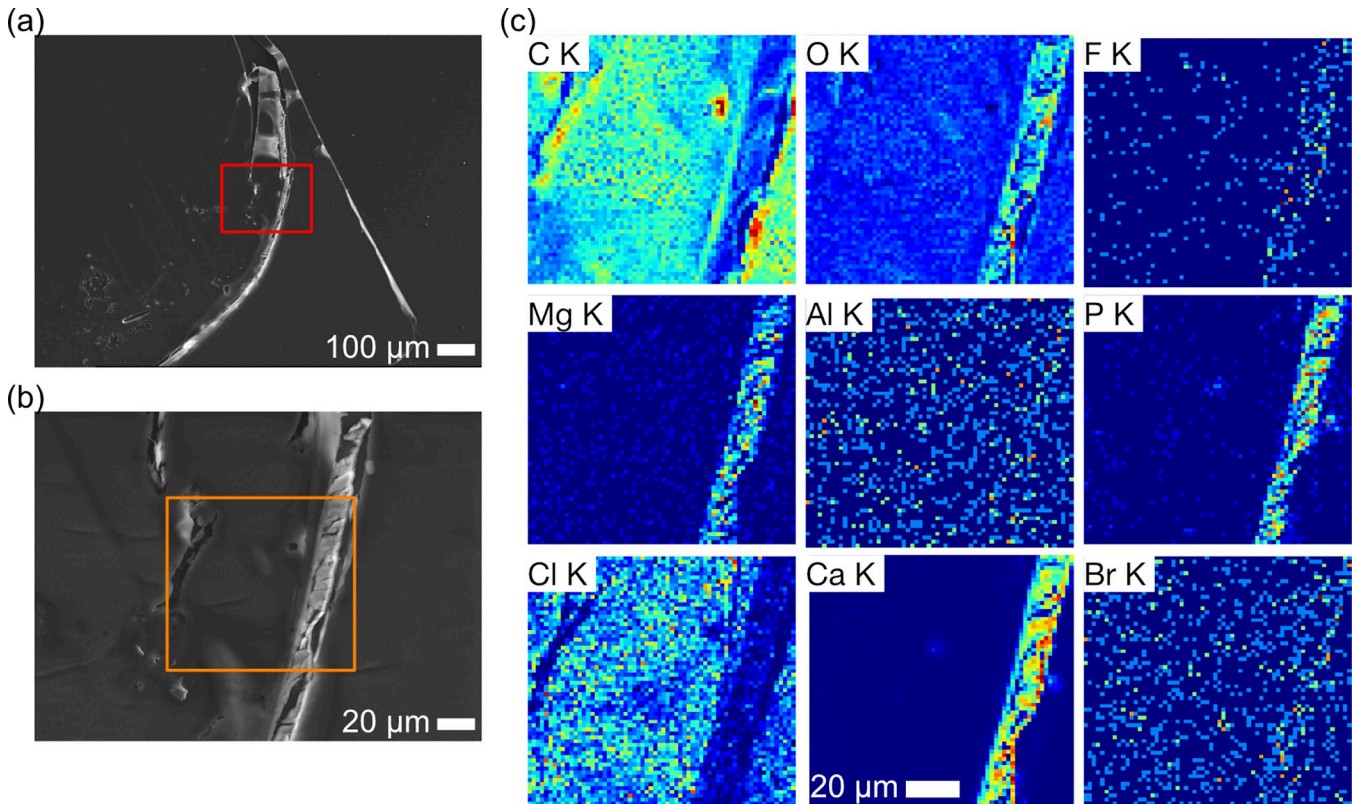

**Fig 4. SEM images and elemental maps of a semi-thin section of the pereonite in *Hirondellea gigas*.** a) Low-magnification SEM image. b) High-magnification SEM image of the red rectangle in (a). c) WDS elemental maps of the orange rectangle in (b).

mill showed two peaks around 1.48 keV: 1.481 and 1.527 keV (Fig 3B top). Aluminum and bromine are the two possible candidates for these peaks: L-line of bromine appears at 1.293 and 1.480 kV while K-line of aluminum appears at 1.486 keV [22]. Quantitative analysis showed that the intensity around 1.48 keV can therefore be explained without aluminum being present, indicating that the aluminum-like signals we have observed could in fact be bromine (S3C Fig). The presence of bromine was further supported by another signal at 11.92 keV which was also observed, corresponding to the K-line of bromine (Fig 3B bottom). EPMA maps showed directional bromine localization toward the side exposed to seawater, similar to that observed in the EDS maps.

The elemental colocalization was further confirmed by calculating cross-correlation factor of the whole image (S1 Table) [26]. A moderate to strong correlation was observed between Ca-O-F-Mg-P, with correlation values ranging from 0.40 to 0.84. On the other hand, the correlation values of bromine against calcium-colocalizing elements were less than 0.1, indicating that the bromine layer is separated from the calcium-based minerals. Quantitative X-ray analysis showed the absence of calcium on the bromine-rich outer layer (S3C Fig). The low concentration of bromine and the coexistence of fluorine and magnesium were also supported by the quantitative analysis (S3 Fig).

Similarly, EPMA maps of the pereonite were taken to explore elements of the exoskeleton (Fig 4, S2 Table). Cross-correlation values indicated the high colocalization of Ca-O-Mg-P with correlation values ranging from 0.15 to 0.70. The correlation of calcium-colocalizing minerals was small due to low X-ray counts. The correlation of bromine to calcium-colocalizing

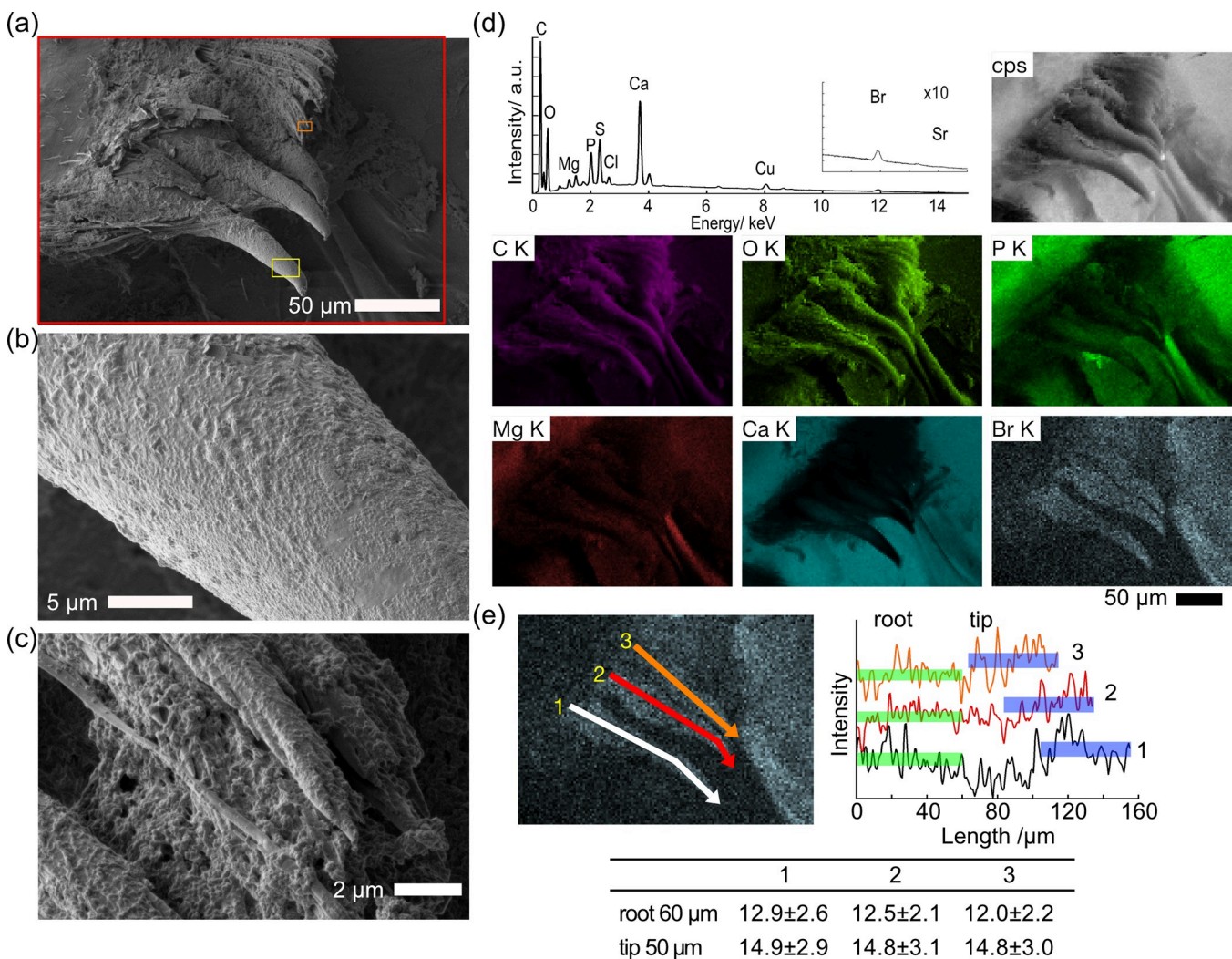

**Fig 5. SEM/EDS images of the gastric mill of *Hirondellea gigas*.** a) Low-magnification SEM image. b) Magnified image of the spine, yellow rectangle in part (a). c) Magnified image of the filter setae, orange rectangle in part (a), d) EDS elemental mapping and sum spectrum of the red rectangle in part (a). e) Line profile of the Br K intensity in EDS map along with the longitudinal axis of the ossicle spines. Profiles were shifted to the intensity direction for visibility, and average ± standard deviation of the intensity of 60 μm from root and 50 μm from the tip are shown in green and blue bars, respectively. The line profile was measured with the line width of 10 μm.

minerals was less than 0.1 except for oxygen, denying strong bromine colocalization. Quantitative analysis clearly showed the absence of either bromine or aluminum (S3B and S3C Fig).

## EDS investigation of the dissected specimen

To further explore the localization of bromine in *Hirondellea gigas*, a dissected individual was freeze-dried and observed under SEM/EDS. EDS maps showed that the hard tissues in *H. gigas* are mainly composed of carbon, nitrogen, oxygen, phosphorous, and calcium as major elements, and fluorine, sodium, magnesium, chlorine, and bromine as minor elements.

Since bromine accumulation was detected on the sclerotized spines of the gastric mill by EPMA, we first examined the gastric mill. The gastric mill exhibited several large, sclerotized spines and numerous fine filter setae, the latter not well observed in thin sections (Figs 5, S4A and S4B). The dimensions of the spines were found to be typically 100 μm in length and 15–

25 μm in diameter (Figs 5A, 5B, S4C and S4E). Fine filter setae around the sclerotized spines were simple and covered by adsorbents, presumably mucus (Figs 5C, S4D and S4F). Elemental maps of Br K indicated an accumulation of bromine on the spines (Fig 5D, S3 Table). Line profile analysis of the Br K line showed that the bromine intensity was 12–13 at the root and 15 at the tip, indicating preferential bromine accumulation at the tip (Fig 5E). The presence of weak calcium signals, combined with EPMA results (Fig 3), point to a bromine-rich and calcium-poor organic layer coating a calcium-rich inorganic layer. Although the intensity was lower, bromine accumulation was also observed on the filter setae (S4H Fig). The pterocardiac ossicle itself at the base of the spines is made of carbon, oxygen, magnesium, phosphorous, and calcium, while bromine was not observed in significant concentrations (S4H Fig).

As fine setae on the gastric mill also accumulated bromine, mouth parts around the gastric mill with fine setae, namely maxilla 2, maxilliped, and mandible, were analyzed by SEM/EDS (Figs 6 and S5–S7). Bromine accumulation was also observed on the setae of these mouth parts, specifically the robust setae. Bromine accumulation was clearest on maxilla 2 (Figs 6A, 6B and S5). The maxilla itself was mainly composed of calcium carbonate together with magnesium and phosphorous, and the bristles were found on the surface, where they are exposed to seawater (Figs 6C, S5E and S5F). Three types of setae were present on maxilla 2 (Fig 6A and 6B). One was simple robust setae with a diameter of 10–20 μm and a length of 150–350 μm, occurring on the distal part of maxilla 2 (S5B Fig right); a second type of robust setae was similar in size but plumose with many fine setae (diameter of 0.6–2.5 μm and length of 35–50 μm, S5B Fig bottom); a third type was fine, setulose setae 0.5–1.5 μm in diameter with a length of 50–150 μm (S5B top, S6C and S6D Figs). EDS maps showed that bromine was accumulated on both types of robust setae with carbon, nitrogen, oxygen, and sulfur, but not on the finer, setulose setae which were rich in phosphorous and calcium (Fig 6C, S3 Table). Similarly, selective bromine accumulation on the robust setae was found in both the maxilliped and the mandible

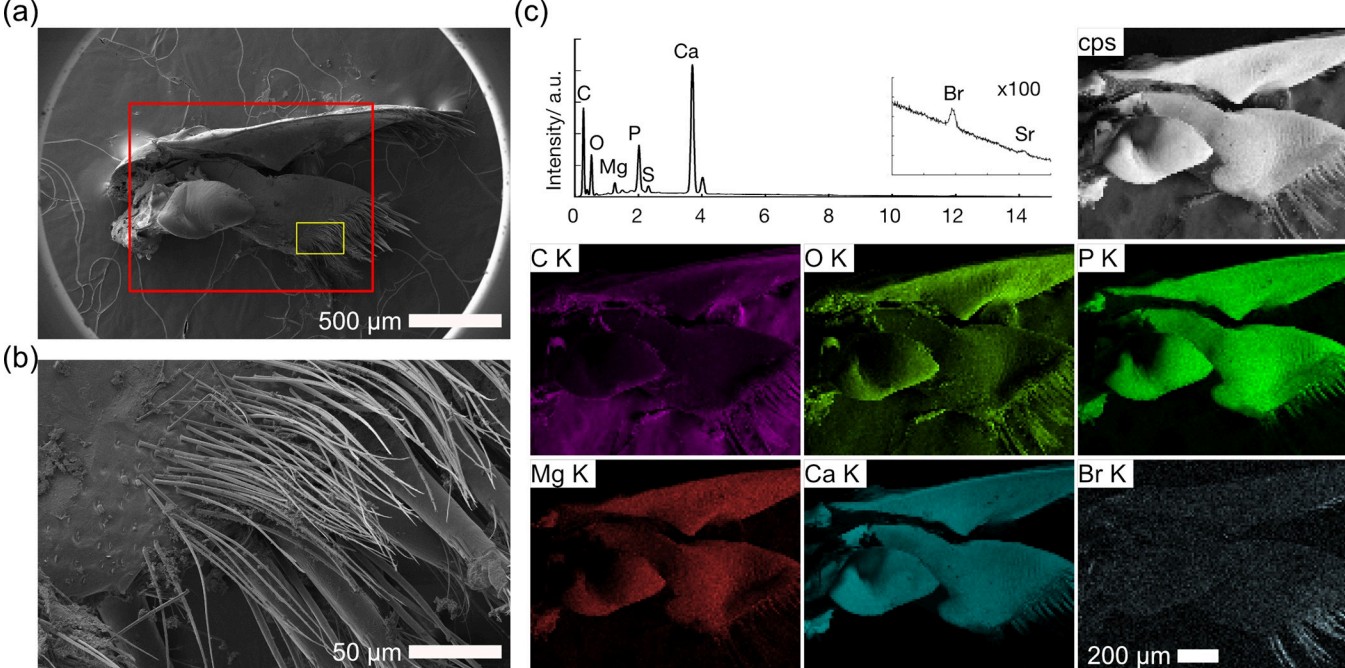

**Fig 6. SEM/EDS images of maxilla 2 of *Hirondellea gigas*.** a) Low-magnification SEM image. b) Magnified image of the yellow rectangle in part (a). c) EDS elemental mapping and sum spectrum of the red rectangle in part (a).

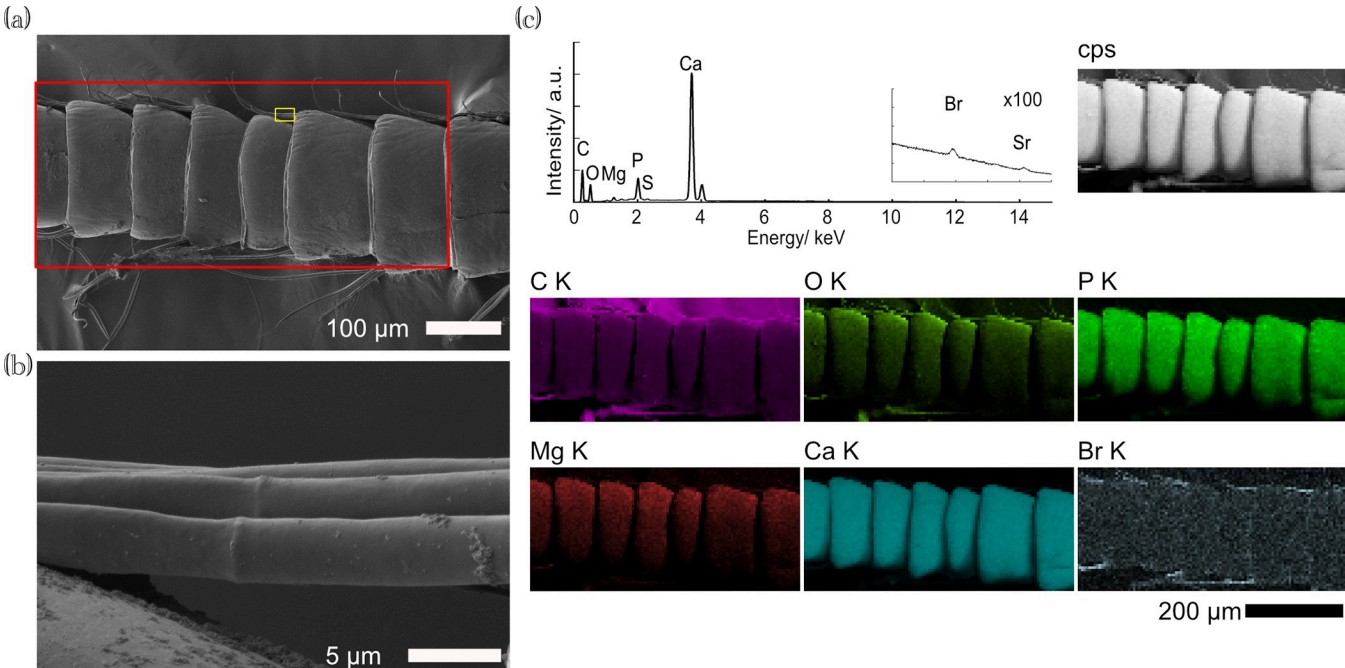

**Fig 7. SEM/EDS images of an antenna of *Hirondellea gigas*.** a) Low-magnification SEM image. b) Magnified image of the yellow rectangle in part (a). c) EDS elemental mapping and sum spectrum of the red rectangle in part (a).

(S6 and S7 Figs, S3 Table). Since the tissues were not washed excessively in order to prevent damage to potentially fragile setae, sodium chloride particles were observed between the setae of maxilliped, and bromine was not observed on the particles (S6C and S6E Figs).

These accumulations of bromine raised the possibility that bromine is widely present on setae across the whole amphipod, and we therefore proceeded to investigate other parts of the body where setae are present, including antenna, pleopods, and uropods. Antenna, fine parts most close to the mouth, (Figs 7 and S8A) also contained the presence of bromine specifically on the setae extending from the distal parts of each article. These setae were simple and fine, 50–80 μm in length and 1–3 μm in diameter (Figs 7A, 7B, S8B and S8D). Setae on the distal end of article 1 of the antennae are, however, barbed and have short barbs projecting out laterally (S8C and S8E Fig). Elemental maps and point analysis revealed bromine accumulation on the setae, free of magnesium, phosphorous, and calcium that were found on the articles themselves (Fig 7C, S3 Table). The setae were characterized by high carbon, nitrogen, oxygen, and sulfur contents, while magnesium and phosphorous were less significant.

Microscopic observations of pleopods revealed bromine accumulation on robust setae as well as the outer surface of the pleopods themselves (Figs 8, S9 and S10). Two types of setae were present, with the robust setae being simple and approximately 4–5 μm in diameter, and elongate fine setae being approximately 0.3–1.2 μm in diameter and about 3 mm in length (Figs 8A, 8B, S9B and S9C). Bromine was found to be accumulated only on the robust setae (Figs 8C and S9D). Unlike other parts of the body, some bromine accumulation was also detected on the surface of the pleopods (Figs 8C and S10, S3 Table).

Uropods possess short distolateral robust setae (Figs 9 and S11A), and we also found bromine accumulating on the setae, which were approximately 10–15 μm in diameter and 80–100 μm in length (Figs 9B, S11B and S11C). The overall surface of the uropods including the setae, lacked bromine accumulation, unlike in the pleopods (Figs 9C and S11D, S3 Table).

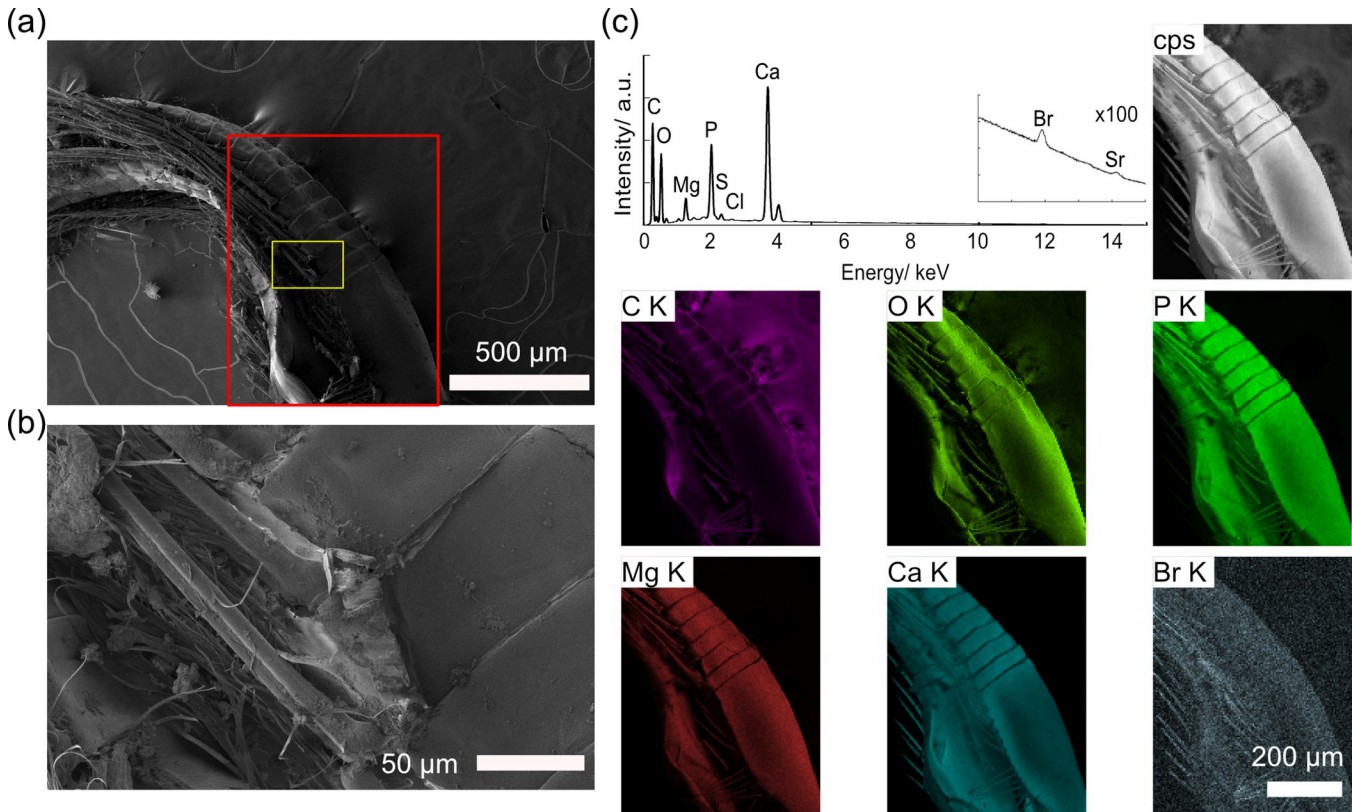

**Fig 8. SEM/EDS images of a pleopod of *Hirondellea gigas*.** a) Low-magnification SEM image. b) Magnified image of the yellow rectangle in part (a). c) EDS elemental mapping and sum spectrum of the red rectangle in part (a).

Since EPMA investigation showed an absence of bromine on the pereonite, suggesting that bromine could be absent on the surface of the dorsal armature of *H. gigas*, we therefore performed SEM/EDS analysis on the coxal plates and pereonites (Figs 10 and S12A). Both coxal plates and pereonites were composed of hexagonal elements approximately 30 μm in size, which were more clearly visible on the coxal plates (Figs 10B and S12C). High-magnification images of the dorsal armature revealed numerous bead-like structures arranged in nearly parallel lines transversing the body, with the typical spacing between the lines being $669 \pm 184$ nm ($N = 36$, S12D Fig). Elemental mapping showed a lack of any specific localization of bromine related to the surface structure in these parts of the exoskeleton, as was also indicated in area analyses (Figs 8C and S12E, S3 Table); although the sum spectra indicated small amounts of bromine.

## Discussion

### Elemental analysis using electron microscopy

We took several precautions during sample preparation to avoid artificial X-ray signals from aluminum outside the sample, which is a very common element and contaminant in electron microscopy [23]. The use of Kapton/carbon reduces the emission of BSE from the substrate, preventing secondary electrons from the SEM chamber (SE3) as well as X-rays from the chamber made of aluminum alloy [21]. The amphipod tissue was mounted as carbon-coated flat semi-thin sections (unlike the bulk specimen in ref. [20]) due to three benefits in X-ray analysis. Firstly, the spatial resolution of a semi-thin section is higher than that of a bulk specimen

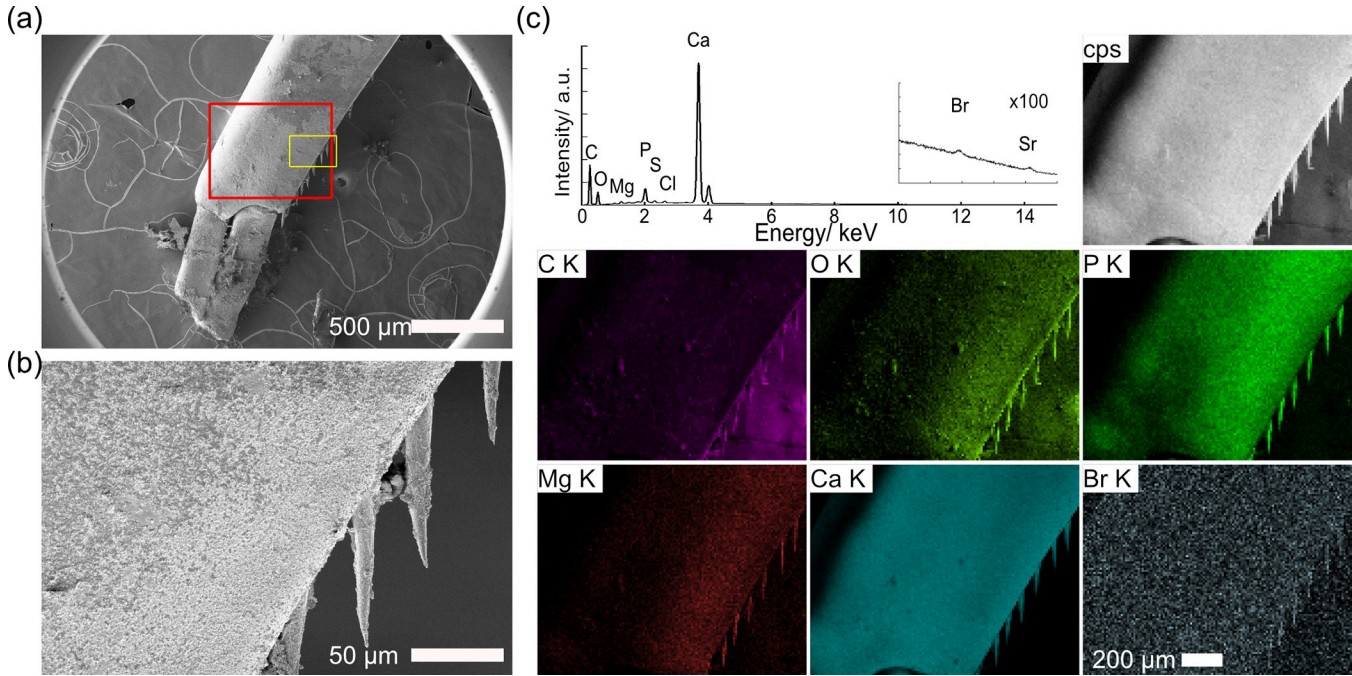

**Fig 9. SEM images and EDS maps of an uropod of *Hirondellea gigas*.** a) Low-magnification image. b) High-magnification image of the yellow square in (a). c) Sum spectrum and EDS maps of the red square in (a). The sum spectra at 10–15 kV were magnified and displayed in the inset above with the same energy scale for clarity.

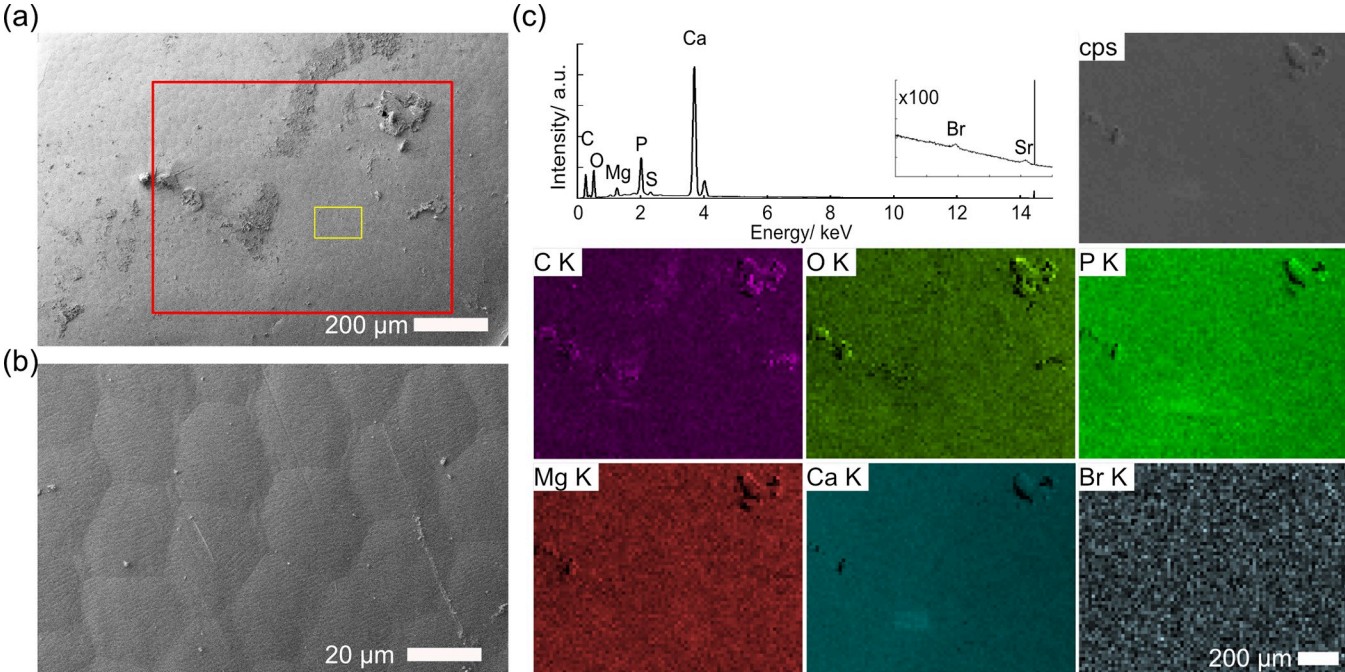

**Fig 10. SEM/EDS images of a coxal plate of *Hirondellea gigas*.** a) Low-magnification SEM image. b) Magnified image of the yellow rectangle in part (a). c) EDS elemental mapping and sum spectrum of the red rectangle in part (a).

because X-ray generation distant from the irradiated position by electrons scattered inside the sample is suppressed. Secondly, the charge-up of the thin section is minimized, as primary electrons irradiated to the thin section are not retained in the sample, but instead penetrate through to the underlying layer, where they are neutralized by the conductive coatings [30]. In the case of bulk samples with an insulative interior, as is the case with biological materials, accumulated charges bend the electron trajectory to decrease spatial resolution. Finally, using semi-thin sections ensure the electrons and X-rays are emitted homogeneously from a flat surface, which is desirable for quantitative analysis. Surface roughness causes shading of X-rays to decrease signal intensities of low-energy X-rays [31]. Without these precautions, total X-ray counts from each position do not necessarily reflect the sample's true elemental composition. We used the counts per second (cps) images, corresponding to total X-ray counts from each pixel, as our criteria to distinguish whether the signal originated from the sample or bremsstrahlung.

After performing the analyses with these precautions, our results showed that what was previously thought to be aluminum signals from *Hirondellea gigas* [20] was most likely bromine. This misassignment of elements can be traced back to issues with the sample preparation and selection of electron microscopic imaging conditions in Kobayashi *et al.*[20]. Although high-resolution imaging can be performed without any coatings by using low-landing voltage techniques under low beam current [32,33], EDS measurements typically require doubled beam energy, i.e. voltage, of the characteristic X-ray energy of the elements analyzed for detection. In this situation, a large beam current is required for X-ray emission sufficient for elemental mapping. The spectrum acquisition condition is prone to cause charge-up, as electrons can penetrate deep into the insulative organic specimen. In addition, internalized high-energy electrons can escape from the side of curved surfaces, resulting in enhanced counts of low-angle BSE– thus the total intensity of X-rays from the SEM chamber and substrate at the sample edge, which corresponds to the previous results where aluminum signals were mostly detected on the sample edges (Fig 1 in ref. [20]). The use of aluminum stubs and aluminum-reinforced double-sided carbon tape, as in ref. [20], is not recommended for aluminum detection for these reasons. Silicon is also an element to be used with caution, because the EDS detector window is made of silicon, which can be excited either by BSE or incoming high-energy X-rays, the latter of which is known as the escape peak.

One criterion to distinguish whether the EDS signals in map images truly originate from elemental localization or are artefacts, is to compare the image with total X-ray counts at each pixel, or counts per second (cps) images. Uncorrected X-ray maps display X-ray intensities of a given X-ray energy range at each position, including bremsstrahlung X-ray that are in correlation with the total emitted X-rays. When the relative intensities of X-ray maps and the cps map of the area concerned are similar in intensity ratio, the element localization is in doubt. Note that conventional X-ray corrections, such as ZAF correction using atom number (Z), X-ray absorption (A) and fluorescence excitation (F) are not necessarily accurate for EDS analysis of non-flat specimens. Increasing the beam energy is another method to distinguish among elements. In the particular case of the aluminum K-line (1.486 keV), the bromine L-line (1.480 keV) and M-lines of two lanthanides absent in animals (thulium and ytterbium) [24] can overlap with it at the energy resolution of EDS (typically ~130 eV). By using high-energy electron beams, the bromine K-line (11.92 keV) can be detected without overlapping with other elements. High-energy X-rays have lower intensities than low-energy ones, and the use of high-sensitivity detectors is desired. The use of semi-thin sections, comparison of X-ray maps and cps maps, irradiation of high-energy electron beam, and the use of the high-sensitivity detector in our analytical methods ensured that we were likely to avoid such artifact pitfalls.

In our measurements, sea salt sometimes appeared on the surface, which is an artefact of the drying process, because we did not wash the tissues thoroughly to prevent removal of fragile structures. However, sea salt only appeared as granules, not as surface coatings, judging from the amount of Na and Cl being far lower than that of Br on fine setae (S4–S8 Figs). This indicates that salt did not affect the amount of bromine because the molar amount of bromine is less than 1% of that of sodium, the main content of the sea salt [34].

## Origin of bromine and its biological implication

Bromine is the eighth most abundant element in the ocean after chlorine, sodium, magnesium, sulfur, calcium, potassium, and carbon [34]. Although the explicit amount of bromine in the Izu-Ogasawara Trench has not been reported [35], the relative amount of bromine is expected to be similar to that of standard seawater. The accumulation of bromine could be a result of 1) solidification and concentration from sea salt during sample preparation, 2) exogenous adhesion on the amphipod's exoskeleton while the amphipod was still alive, or 3) active accumulation and secretion by the amphipod itself. We observed solidified sea salt on the setae of maxilliped, but these were without bromine coexistence (S6C–S6E Fig), rejecting the bromine signals being results of artefacts from sample preparation. Exogenous adhesion is also considered to be unlikely since chlorine, which has similar chemical properties as a water-dissolved anion and being more abundant in seawater than bromine, was absent on the setae. Rather, it is more plausible to consider that bromine was secreted by the amphipod, most preferentially to the gastric mill but also to the robust setae on various parts of the body, as well as the surface of the pleopods.

Accumulation of bromine has also been found on other marine organisms, and bromine-rich aromatic small molecules have been isolated from marine organisms. For example, the marine snail *Bolinus bandaris* (a historically important source of Tyrian purple) secretes 6-bromo-3-indolinone derivatives [36], the tripeptide barettin from the marine sponge *Geodia barretti* has dehydrogenated 6-bromotryptophan[37], and marine bacteria such as *Pseudomonas bromoutilis* produce pentabrominated phenylpyrrole, which is an antibiotic compound [38,39]. Bromine is attached on aromatic rings in all cases. There are also examples of bromine being accumulated on the hard parts of marine organisms whose molecular structure has not been identified, such as the fish otolith containing bromine and chlorine colocalized as minor elements [40]. Such cases of bromine in hard structures are also known from at least two cases where the hard parts have predation or feeding-related functions. In the crab *Pachygrapsus crassipes*, bromine is enriched on the tips of its calcified claws, and mechanical tests have shown that this makes the tips less hard, yet more fracture-resistant [16,17]. Another case is the zinc-based jaws of the marine polychaete worm *Nereis virens* that also contain chlorine, manganese, and iodine[2,3]. In this polychaete worm, bromine is found as brominated amino acids, such as histidine, which likely contributes to preventing dissolution by binding strongly to metals [2,3].

The hadal amphipod *H. gigas* is unique among these animals in that the bromine does not coexist with other halogens and it is poor in other transition metals within the 1-μm thick organic coatings containing bromine. In *H. gigas*, bromine is expected to reinforce the gastric mill to crush food in the same manner as reinforcement occurs in the jaws of other animals. The localization of bromine on the sclerotized spines and the lack of it in the calcified ossicles itself is likely beneficial in that bromine renders the spines less susceptible to breaking, in order to maintain its sharpness and function in food sorting. The ossicles are used for actual crushing, and as such it is beneficial for them to be calcified to increase their hardness. The same has been suggested for the function of the forceps-like, bromine-enriched, crab claw tips

(which is calcified and lacks bromine) versus the rest of the claw used for crushing [16]. The form of bromine in the gastric mill and setae remains unclear. The coexisting nitrogen and sulfur with bromine indicate the presence of amino acids, which may imply that the bromine is incorporated in aromatic amino acids, such as phenylalanine and tryptophan, in which case the bromine is most likely secreted by *H. gigas*. The function of bromine on the setae of *H. gigas* remains unclear at this point, although it may be secreted in a similar manner to the spines in the gastric ossicles.

Calcite as the exoskeleton at depths deeper than the CCD can dissolve, and how *H. gigas* is able to retain its calcite armature in such conditions requires reconsideration now that it has become clear that it does not possess an 'aluminum shield' as previously suggested [20]. Our EDS and XRD results agrees with the previous observation [20] that the exoskeleton is composed of calcite and proteinaceous organic materials (Figs 2A, S1A and S2). EPMA showed the existence of magnesium in the calcite layer with the Mg/Ca ratio being around 2–20% (S2 Fig). The absence of chlorine in the dried specimen (S4–S12 Figs) indicates that the magnesium is not from the remaining seawater salt, but is within the exoskeleton. Water solubility of calcite decreases with slight substitution of calcium with magnesium, i.e. $(Ca,Mg)CO_3$ [41,42], and the substitution would partially prevent carbonate dissolution below the CCD. Phosphorous may be present as calcium phosphate, contributing to the stabilization of calcium carbonate minerals [5,6,10].

## Conclusion

In summary, we revealed the presence of bromine and magnesium in the hadal amphipod *Hirondellea gigas*, and showed through careful prevention of aluminum artefacts that it does not actually possess an 'aluminum shield' as previously thought. Instead of the 'aluminum gel' shielding, the use of magnesium-substituted calcite may be the key for the amphipod to survive without demineralization below the CCD. High concentrations of bromine were found on the sclerotized spines in the gastric mill, forming a 1 μm-thick coated layer on mineralized tissue with a relatively high concentration of nitrogen. The bromine-coating on the gastric mill spines is speculated to have the function of maintaining a sharp tip for food processing, similar to what has been suggested for crab claw tips. Our study serves as a reminder of the importance and necessity of appropriate sample preparation and careful analysis of the signals in EDS analyses, as well as the importance of analyzing samples at different electron energies when performing microscopic studies of 'elemental anatomy'.

## Supporting information

**S1 Fig. EDS maps of semi-thin sections of *H. gigas*.** a) Transverse section through a subsample of pereonite. b) Transverse section of the cephalic region showing the gastric mill. c) Magnified images of the gastric mill, the red rectangle area in (b). SEM images, total X-ray counts (cps), and elemental maps are shown, and the labels are denoted in the left top. 'Al' corresponds to the aluminum-like signal around 1.48 keV, that overlaps with bromine L-line, and thulium and ytterbium M-line. d–f) EDS sum spectra of the (d) pereonite, (e) cephalic region, and (f) gastric mill created from the EDS maps (a)–(c). SEM images in (a) were acquired at 1 kV, and other images and spectra were acquired at 10 kV.
(TIF)

**S2 Fig. XRD spectrum of dried pereonite of *H. gigas*.** Diffraction peak position in 2θ and corresponding crystal planes of calcite and α-chitin in parenthesis are shown.
(TIF)

**S3 Fig. Quantitative point analysis of EPMA using semi-thin sections.** a) SEM images of the gastric mill. b) SEM image of the pereonite. c) Atomic ratio of the elements. K-line of the elements was used for the quantification. The large value of carbon and oxygen is ascribed to the conductive carbon coating and the underlying Kapton tape. Note that the amount of aluminum was <5% of that of bromine, and may contain some error because of the overlap of Al K-line with Br L-line, which is not completely separable under WDS conditions (Fig 2B). Nitrogen was also searched but not observed at any points (0.00), thus not shown.
(TIF)

**S4 Fig. SEM images and EDS spectra of the gastric mill of *Hirondellea gigas*.** a) Low-magnification SEM image. b) Magnified image of the red rectangle in (a). c) Magnified image of the orange rectangle in b). d) Magnified image of the blue rectangle in (b). e) Magnified image of the yellow rectangle in (c). f) Magnified image of the green rectangle in (d). g) EDS elemental mapping of the red rectangle in (a). h) EDS spectra of the yellow points in (a). Atom labels correspond to K-line except for Br*, which is L-line.
(TIF)

**S5 Fig. SEM images and EDS spectra of the maxilla 2 of *Hirondellea gigas*.** a) Low-magnification SEM image. b) Magnified image of the red rectangle in (a). c) Magnified image of the orange rectangle in b). d) Magnified image of the yellow rectangle in (c). e) EDS elemental mapping of the blue rectangle in (a). f) EDS spectra of the yellow points in (b). Atom labels correspond to K-line except for Br*, which is L-line.
(TIF)

**S6 Fig. SEM images and EDS spectra of the maxilliped of *Hirondellea gigas*.** a) Low-magnification SEM image. b) Magnified image of the red rectangle in (a). c) Magnified image of the orange rectangle in b). Note that the solid between setae are unremoved sodium chloride, assigned based on EDS mapping in (e). d) EDS spectra of the yellow points in (b). e) EDS elemental mapping of the blue rectangle in (a). Atom labels correspond to K-line except for Br*, which is L-line.
(TIF)

**S7 Fig. SEM images and EDS spectra of the mandible of *Hirondellea gigas*.** a) Low-magnification SEM image. b) Magnified image of the red rectangle in (a), c) Magnified image of the orange rectangle in b). d) Magnified image of the yellow rectangle in (c). e) EDS maps of the blue rectangle in (a). f) EDS spectra of the yellow points in (b). Atom labels correspond to K-line except for Br*, which is L-line.
(TIF)

**S8 Fig. SEM images and EDS spectra of the antenna of *Hirondellea gigas*.** a) Low-magnification SEM image. b) Magnified image of the red rectangle in (a). c) Magnified image of setae from the article in the yellow rectangle in (b). d) Magnified image of the orange rectangle in (a). e) Magnified image of short barbs in the green rectangle in (c). f) EDS elemental mapping of the blue rectangle in (a). g) EDS spectra of the yellow points in (a). Atom labels correspond to K-line except for Br*, which is L-line.
(TIF)

**S9 Fig. SEM images and EDS spectra of the pleopod of *Hirondellea gigas*.** a) low-magnification SEM montage image. b) Magnified image of the red rectangle near the red triangle in (a) showing the two types of setae. c) Magnified image of the yellow rectangle in b). d) EDS spectra of the yellow points in (a). Atom labels correspond to K-line except for Br*, which is L-line.
(TIF)

**S10 Fig. SEM/EDS mapping of the pleopod of *Hirondellea gigas*.** a) Low-magnification SEM montage image, which is a part of S8 Fig. b) EDS elemental mapping of the blue rectangle in (a).
(TIF)

**S11 Fig. SEM images and EDS spectra of the uropod of *Hirondellea gigas*.** a) Low-magnification SEM image. b) Magnified image of the red rectangle, left middle in (a). c) Magnified image of the orange rectangle in b). d) EDS spectra of the yellow points in (b). e) EDS elemental mapping of the blue rectangle in (a). Atom labels correspond to K-line except for Br*, which is L-line.
(TIF)

**S12 Fig. SEM images and EDS spectra of the pereonite of *Hirondellea gigas*.** a) Low-magnification SEM image, b) Magnified image of the red rectangle, left middle in (a). c) Magnified image of the orange rectangle in (b) showing the hexagonal elements. d) Magnified image of (b) showing the bead-like structure. e) EDS sum spectra of (e). The surface was elementally homogeneous and the EDS maps are not shown. Atom labels correspond to K-line except for Br*, which is L-line.
(TIF)

**S1 Table. Cross correlation of EPMA elemental maps of semi-thin section of the pereonite in Fig 2.** Top 10 values are shown in bold for visibility.
(PDF)

**S2 Table. Cross correlation of EPMA elemental maps of semi-thin section of the gastric mill in Fig 3.** Top 10 values are shown in bold for visibility.
(PDF)

**S3 Table. Elemental ratios of EDS point/area analyses in S4–S12 Figs.** GM, gastric mill; Mxp, maxillipet; Md, mandible; Mx2, maxilla 2; An, antenna; Pl, pleopod; Ur, uropod; Cx, coxal plate; Pe, pereonite. Values are reported by the atomic ratio ± errors in per cent. Note that the spectra were acquired under non-optimal condition, i.e. curved surface, and may contain large errors in the relative values. Area analysis was performed on Cx and Pe.
(PDF)

## Acknowledgments

The authors thank the Captain and crew of R/V *Yokosuka* during expeditions YK19-11 (cruise PI: Akinori Yabuki, JAMSTEC) for their great support of the scientific activity. All scientists on-board the expeditions are gratefully acknowledged for their tireless work, especially Heather Ritchie Parker who assisted with the preservation of the amphipod specimens. We thank Kana Nagashima (JAMSTEC) for her kind offer for XRD analysis. We thank three anonymous reviewers for providing useful comments that improved earlier versions of this paper.

## Author Contributions

**Conceptualization:** Satoshi Okada, Chong Chen, Ken Takai.

**Data curation:** Satoshi Okada, Hiromi Kayama Watanabe, Noriyuki Isobe.

**Formal analysis:** Satoshi Okada, Noriyuki Isobe.

**Funding acquisition:** Satoshi Okada, Chong Chen, Hiromi Kayama Watanabe.

**Investigation:** Satoshi Okada, Chong Chen, Hiromi Kayama Watanabe, Noriyuki Isobe.

**Methodology:** Satoshi Okada, Chong Chen, Hiromi Kayama Watanabe.

**Project administration:** Ken Takai.

**Supervision:** Ken Takai.

**Validation:** Hiromi Kayama Watanabe, Noriyuki Isobe, Ken Takai.

**Visualization:** Satoshi Okada.

**Writing – original draft:** Satoshi Okada.

**Writing – review & editing:** Chong Chen, Hiromi Kayama Watanabe, Noriyuki Isobe, Ken Takai.

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
