## [Decision Letter · Decision Letter 0]

7 Jun 2022

PONE-D-22-02007Unusual bromine enrichment in the gastric mill and setae of the hadal amphipod Hirondellea gigasPLOS ONE

Dear Dr. Okada,

Thank you for submitting your manuscript to PLOS ONE. After careful consideration, we feel that it has merit but does not fully meet PLOS ONE’s publication criteria as it currently stands. Therefore, we invite you to submit a revised version of the manuscript that addresses the points raised during the review process. Regarding the reviews, I would just like to add that I personally have no concerns about a large amount of supporting information provided with the manuscript. I welcome if as much as possible of the pertinent data is provided with a manuscript. 

We look forward to receiving your revised manuscript.

Kind regards,

Hannes C Schniepp, Dr. sc. nat.

Academic Editor

PLOS ONE

Journal Requirements:

Reviewers' comments:

Reviewer's Responses to Questions

**Comments to the Author**

1. Is the manuscript technically sound, and do the data support the conclusions?

Reviewer #1: Yes

Reviewer #2: Yes

2. Has the statistical analysis been performed appropriately and rigorously? 

Reviewer #1: I Don't Know

Reviewer #2: Yes

3. Have the authors made all data underlying the findings in their manuscript fully available?

Reviewer #1: Yes

Reviewer #2: Yes

4. Is the manuscript presented in an intelligible fashion and written in standard English?

Reviewer #1: Yes

Reviewer #2: No

5. Review Comments to the Author

Reviewer #1: I read your paper with great interest. The authors used detailed image analysis by electron microscopy and X-ray measurements on amphipod samples collected from the difficult-to-reach ultra-deep sea, and found that amphipods do not have an "aluminum shield" as previously thought, but accumulate bromine and magnesium. We believe that these new findings will greatly contribute to the development of deep-sea biology. This paper is judged to be of sufficient value for publication in PlosOne.

Reviewer #2: See attached documents provided.

Review of “Unusual bromine enrichment in the gastric mill and setae of the hadal amphipod Hirondellea gigas” by Okada, et al.

I have reviewed the ms by Okada, et al, regarding the absence of an aluminum gel shield in the deep-water amphipod, H. gigas and the presence of bromine, instead, demonstrating the importance of careful and proper SEM analyses of samples for ‘elemental anatomy.’ I found the manuscript (ms) of interest and of potential interest to the PLOS One audience of readers. However, although the work appears to be thorough and meticulously done, demonstrating careful analytical rigor, I have, nevertheless found the writing to be rather simplistic with faults in the logical flow of the paper. It appears more analogous to an undergraduate laboratory report, than a ms for publication, and for the most part, is poorly prepared.

6. PLOS authors have the option to publish the peer review history of their article (what does this mean?). If published, this will include your full peer review and any attached files.

Reviewer #1: No

Reviewer #2: No

---

## [Author Response · Author response to Decision Letter 0]

13 Jun 2022

Thank you very much for Reviewer 2's critical comments, which were extremely helpful to revise our manuscript. The manuscript was revised accordingly, and the point-to-point response is shown in the separate file.

---

## [Decision Letter · Decision Letter 1]

8 Jul 2022

PONE-D-22-02007R1Unusual bromine enrichment in the gastric mill and setae of the hadal amphipod Hirondellea gigasPLOS ONE

Dear Dr. Okada,

Thank you for submitting your manuscript to PLOS ONE. After careful consideration, we feel that it has merit but does not fully meet PLOS ONE’s publication criteria as it currently stands. Therefore, we invite you to submit a revised version of the manuscript that addresses the points raised during the review process. Specific Editor Comment: Please address the feedback from both reviewers regarding the minor issues mentioned.

We look forward to receiving your revised manuscript.

Kind regards,

Hannes C Schniepp, Dr. sc. nat.

Academic Editor

PLOS ONE

Journal Requirements:

Reviewers' comments:

Reviewer's Responses to Questions

**Comments to the Author**

1. If the authors have adequately addressed your comments raised in a previous round of review and you feel that this manuscript is now acceptable for publication, you may indicate that here to bypass the “Comments to the Author” section, enter your conflict of interest statement in the “Confidential to Editor” section, and submit your "Accept" recommendation.

Reviewer #2: (No Response)

Reviewer #3: (No Response)

2. Is the manuscript technically sound, and do the data support the conclusions?

Reviewer #2: Yes

Reviewer #3: Yes

3. Has the statistical analysis been performed appropriately and rigorously? 

Reviewer #2: Yes

Reviewer #3: Yes

4. Have the authors made all data underlying the findings in their manuscript fully available?

Reviewer #2: Yes

Reviewer #3: Yes

5. Is the manuscript presented in an intelligible fashion and written in standard English?

Reviewer #2: No

Reviewer #3: No

6. Review Comments to the Author

Reviewer #2: The ms is VERY MUCH improved in readability and logical flow. I congratulate the authors on really paying attention to the review comments I provided in the original review. The ms needs now only minor grammatical corrections, for which I have included suggestions and edits in the PDF of the revised text of ms attached.

Reviewer #3: The authors carefully examined the minor element composition of the hadal amphipod and found that the amphipod showed bromine enrichment in the gastric mill and setae, but no evidence for Al enrichment. This is my first evaluation of this MS, but I took over the review comments provided by earlier reviewers. The authors properly answered the reviewer's comments. I also felt that the manuscript is properly written and the scientific quality of the manuscript meets the standard of Plos One publication.

However, there are still some mistakes in the manuscript that should be corrected before publication, even though the reviewer pointed out the English editing. I only listed the mistakes I could find, but I strongly recommend that all of the authors carefully multiple-check the manuscript, given the Plos One policy is that "PLOS ONE does not copyedit accepted manuscripts, so the language in submitted articles must be clear, correct, and unambiguous. Any typographical or grammatical errors should be corrected at revision". English editing is not the job of a reviewer. The authors should English editing and proofreading with their own responsibility.

L20, should be corrected properly.

L80, should be corrected properly.

L131, Please explain what is QY-1.

L132, laser should be razor?

7. PLOS authors have the option to publish the peer review history of their article (what does this mean?). If published, this will include your full peer review and any attached files.

Reviewer #2: No

Reviewer #3: No

---

## [Author Response · Author response to Decision Letter 1]

11 Jul 2022

Thank you for the revision. We corrected the manuscript accordingly, and the point-by-point response is attached as a separate file.

---

## [Editor Report · Decision Letter 2]

13 Jul 2022

Unusual bromine enrichment in the gastric mill and setae of the hadal amphipod Hirondellea gigas

PONE-D-22-02007R2

Dear Dr. Okada,

We’re pleased to inform you that your manuscript has been judged scientifically suitable for publication and will be formally accepted for publication once it meets all outstanding technical requirements.

Kind regards,

Hannes C Schniepp, Dr. sc. nat.

Academic Editor

PLOS ONE
---

## [Editor Report · Acceptance letter]

15 Jul 2022

PONE-D-22-02007R2 

Unusual bromine enrichment in the gastric mill and setae of the hadal amphipod *Hirondellea gigas*

Dear Dr. Okada:

I'm pleased to inform you that your manuscript has been deemed suitable for publication in PLOS ONE. Congratulations! Your manuscript is now with our production department. 

Kind regards, 

on behalf of

Dr. Hannes C Schniepp 

Academic Editor

PLOS ONE